Neuropeptides from a praying mantis: what the loss of pyrokinins and tryptopyrokinins suggests about the endocrine functions of these peptides

http://orcid.org/0000-0002-2783-0018 Veenstra Jan A. jan-adrianus.veenstra@u-bordeaux.fr
INCIA UMR 5287 CNRS, Université de Bordeaux , Bordeaux , France
Kent Clement
Electronic publication date: 2025 Feb 27
Publication date: 2025
Volume: 13
Electronic Location ID: e19036
Received 2024 Aug 26; Accepted 2025 Jan 31
Copyright: © 2025 Veenstra
Copyright year: 2025
Copyright holder: Veenstra
License: This is an open access article distributed under the terms of the Creative Commons Attribution License, which permits unrestricted use, distribution, reproduction and adaptation in any medium and for any purpose provided that it is properly attributed. For attribution, the original author(s), title, publication source (PeerJ) and either DOI or URL of the article must be cited.
License URL: https://creativecommons.org/licenses/by/4.0/

Keywords: Pyrokinin, Tryptopyrokinin, Pyrokinin-1 receptor, Salivary gland, Evolution, Function, Praying mantis, Hugin

Funding: The author received no funding for this work.

==============================
Background

Neuropeptides play important roles in insects, but in many cases their functions remain obscure. Comparative neuropeptidome analyses may provide clues to what these functions might be. Praying mantises are predators and close relatives of cockroaches that are scavengers. Cockroach neuropeptidomes are well established, but little is known about mantis neuropeptides. The recently published genome assembly of the praying mantis Tenodera sinensis makes it Possible to change that.

Methods

The genome assembly from T. sinensis was analyzed for the presence of genes coding neuropeptides. For comparison, publicly available short read archives from this and other mantis species were also examined for the presence and expression of neuropeptides.

Results

As a rule, the neuropeptidomes of the Mantodea and Blattodea are almost identical; praying mantises and cockroaches use very similar neuropeptides. However, there is one surprising exception. Praying mantises lack the receptors for pyrokinins, including those for the tryptopyrokinins. No typical pyrokinin genes were found, but some species do have a tryptopyrokinin gene, in others this has also been lost and, in one species it is a speudogene. For most praying mantises there is no information where tryptopyrokinin is expressed, but in Deroplatys truncata it is in the thorax and thus not in the suboesophageal ganglion, as in other insects. In the genomic short read archives of two species–out of 52–sequences were found for a tryptopyrokinin specific receptor. The phylogenetic position of those two species implies that the receptor gene was independently lost on multiple occasions. The loss of the tryptopyrokinin gene also happened more than once.

Discussion

The multiple independent losses of the pyrokinin receptors in mantises suggests that these receptors are irrelevant in praying mantises. This is very surprising, since expression of tryptopyrokinin is very strongly conserved in two neuroendocrine cells in the suboeosphageal ganglion. In those species for which this is known, the expression of its receptor is in the salivary gland. As a neuroendocrine, tryptopyrokinin is unlikely to acutely regulate salivation, which in other insects is regulated by well characterized neurons. If the action of tryptopyrokinin were to prime the salivary gland for subsequent salivation, it would make perfect sense for a praying mantis to lose this capacity, as they can not anticipate when they will catch their next prey. Priming the salivary gland days before it is actually needed would be energetically costly. The other pyrokinins are known to facilitate feeding and may in a similar fashion prime muscles needed for moving to the food source and digesting it. This hypothesis provides a good explanation as to why praying mantises do not need pyrokinins, and also what the function of these ubiquitous arthropod neuropeptides may be.

Introduction

In insects, as in other animals, many physiological processes are regulated by neuropeptides, either at the periphery or within the nervous system. Many of those that are released into the circulation are often well conserved and have identical or very similar functions. In some cases these functions are well defined, as is the case for the various neuropeptides regulating molting (Žitnan & Adams, 2012). For others, these functions are not as clear. For instance, there are a number of neuropeptides that stimulate fluid secretion by the Malpighian tubules, but their primary functions may not necessarily concern the elimination of excess water.

Praying mantises are fascinating insects, as ambush predators that can sit still for hours, biding their time until a suitable prey is close enough to be grabbed by their powerful forelegs. They occasionally eat even small vertebrates, but usually their prey consists of other insects. Sometimes an unlucky male falls prey to its female partner after, or even during copulation. The silhouette of a mantis with its forelegs in position resembles the hands and arms during prayer and led both to its common name, praying mantis, and that of a common European species, Mantis religiosa. Their morphology, physiology and behavior are well adapted to this lifestyle. Many praying mantises have green body color that allows them to perfectly blend into the vegetation, while orchid mantises camouflage as flowers. An ambush predator must be able to rapidly detect and capture possible prey. Indeed, mantises have the best visual detection system in insects, and it allows them three dimensional vision, while their front legs have been modified into a powerful prey capturing and holding machine.

Praying mantises belong to the insect order Mantodea. Although all Manodea are mantises and predators, not all mantises employ an ambush strategy; the metallic mantises actively hunt and use their forelegs not only to catch prey, but also to move around, which the other mantises do not. The Mantodea and the Blattodea, the insect order comprising termites and other cockroaches, are closely related and are the only insect orders belonging to the Dictyoptera, a superorder within the Polyneoptera. The Polyneoptera have much richer neuropeptidomes than the Eumetabola. So far only a few neuropeptides have been reported from the Mantodea (Koehler & Predel, 2010; Gäde & Marco, 2017). Since the mantises are predators, while termites and other cockroaches are scavengers, it would be interesting to see to what extent their neuropeptidomes resemble or diverge from those of of cockroaches, that are well characterized. The recently published genome assembly of the praying mantis Tenodera sinensis (Yuan et al., 2022) provides the opportunity to do so. I here report that although many neuropeptides in this group appear to be very similar to those of other Polyneoptera, and in particular those of the Blattodea, T. sinensis, like many other mantises, lacks pyrokinins, while it also has a few neuropeptide curiosities that seem to be limited to only a few species. The absence of the pyrokinins from these insects provides important insight to their function and illustrates how neuropeptide signaling systems may be lost during evolution.

Materials and Methods

Portions of this manuscript were previously published as part of a preprint (https://www.biorxiv.org/content/10.1101/2024.08.22.609177v1). The Tenodera sinensis genome assembly was downloaded from NCBI (https://www.ncbi.nlm.nih.gov/nuccore/JASJEM000000000). In order to facilitate its analysis, the short read archive (SRA) SRR25309874 that contains short genomic sequences, as well as the PACBIO_SMRT SRA SRR28416710 were also used. Initial searches for neuropeptide genes used the tblastn command of the BLAST+ program (https://ftp.ncbi.nlm.nih.gov/blast/executables/blast+/LATEST/) on the genome assembly, using as search query neuropeptide precursor sequences from other Polyneoptera, mostly those from Periplaneta americana and other cockroaches (Zeng et al., 2021; Veenstra, 2023). For most genes, this yielded partial neuropeptide precursor sequences. Given large intron sizes and the relatively poorly conserved signal peptide sequences, it was usually difficult to obtain complete sequences using this approach. To complete and verify these sequences, RNA sequencing (RNAseq) SRAs (SRR25098587, SRR25098588, SRR25098589, SRR25098590, SRR25098591, SRR25098593 and SRR25098594) were used in combination with the SRAtoolkit (https://github.com/ncbi/sra-tools/wiki/01.-Downloading-SRA-Toolkit) and Trinity (https://github.com/trinityrnaseq/trinityrnaseq/), using methodology explained in detail elsewhere (Veenstra, 2019). In a few instances, this provided insufficient information and the RNAseq SRAs (SRR25249010, SRR25249011, SRR25249012, SRR25249013, SRR25249014, SRR25249015, SRR25249016, SRR25249017, SRR25249018, SRR25249019, SRR25249020, SRR25249021, SRR25249022, SRR25249023 and SRR25249024) from the closely related T. angustipennis (Mishina et al., 2023) were used to find its ortholog in T. angustipennis, which subsequently permitted to finish the T. sinensis coding sequences to completion. In initial searches, three neuropeptide genes, i.e., the EFLamide, pyrokinin and tryptopyrokinin genes, were not detected in the T. sinensis genome assembly.

The reads from a genomic SRA that are produced without polymerase chain reaction (PCR) amplification are random sequences from the genome. When their combined length represent more than 50 times the genome size, the chance that a particular sequence from the genome is completely absent, i.e., is not represented a single time in the SRA, is essentially zero (less than 0.000001). Therefore, if one can reliably detect coding sequences in short sequence, it is possible to state with the same probability that such a sequence is absent from a genome, when they are absent from such SRAs. The transmembrane regions of G-protein coupled receptors (GPCRs) are so well conserved, that they are reliably detected in short sequences and one can use this criterion to demonstrate the absence of specific well conserved GPCRs (cf Veenstra, 2019). If a specific GPCR is absent from a genome, searching for it using the BLAST+ program yields closely related homologs. For neuropeptides that are well conserved, such as e.g., the Drosophila insulin-like peptide 7 (dilp7) orthologs and the various allatostatin Cs, one can use the same method. However, for small neuropeptides, that are not well conserved or consist of only a small number of amino acid residues that have multiple codons, this is often not feasible. In this manuscript this applies to both EFLamide and the pyrokinins. Therefore, when the core sequence of a neuropeptide is small, it is often difficult to detect in a genome assembly, particularly when genomes are as large as in Tenodera. In order to establish that these signaling systems are indeed genuinely absent from this species, I used the coding exons of the EFLamide receptor from Locusta migratoria (Veenstra & Šimo, 2020), and the pyrokinin receptors from the American cockroach, Periplaneta americana (deduced from the genome and publicly available SRAs), as queries to search one of the genomic SRAs (SRR25309874) using the tblastn_vdb command from the SRAtoolkit. The rationale behind this is that the transmembrane regions of these GPCRs are sufficiently conserved to find homologous sequences in such short read genomic SRAs. To provide evidence that the absence of the pyrokinins is not limited to a single species, genomic SRAs from four additional Mantodea, i.e., Deroplatys truncata (SRR25068526), Hymenopus coronatus (SRR25046609), Mantis religiosa (SRR25010894) and Metallyticus violocea (SRR25078554) were analyzed in the same fashion. Like the genomic SRAs from T. sinensis, the genomic SRAs from those four species are also so large, that the probability that the absence of conserved sequences—such as those from the transmembrane regions of GPCRs—could be due to chance, is virtually zero. In order to get a still better idea of the distribution of tryptopyrokinins in Mantodea, a large number of genome and transcriptome SRAs (listed in Tables S1 and S2) from this order were also investigated for the presence of individual reads coding tryptopyrokinins and/or their receptors. All SRAs used in this manuscript can be found on the NCBI website (https://www.ncbi.nlm.nih.gov/sra).

Fasttree (Price, Dehal & Arkin, 2010) was used to create a sequence similarity tree of a collection of insulin-like peptides (ilps) from cockroaches (sequences from Veenstra, 2023) and mantises. The complete deduced coding sequences were aligned using Clustal omega (Sievers et al., 2011).

Results

Almost all known insect neuropeptide precursors could be identified in the T. sinensis genome assembly and the complete precursor sequences predicted, although for a few genes only partial coding sequences were obtained (Fig. S1, Supplemental Spreadsheet). Most of the Tenodera neuropeptide sequences do not seem to be significantly different from those in other Polyneoptera and will not be further discussed, but a few merit special mention.

Initial searches for an EFLamide ortholog were fruitless, but as the Tenodera genome contains an ortholog of the Locusta EFLamide receptor (Fig. S2), a gene coding its ligand should be expected. Extensive searches yielded what is likely the last coding exon of a gene coding SLGSELL-amide, the putative EFLamide ortholog. A similar exon was also identified in other mantises. Interestingly, an ortholog for the EFLamide receptor is lacking from the American, but present in the German cockroach, which also has a coding exon of a similar EFLamide related peptide (Fig. 1). Low expression levels of these peptides did not allow for the identification of the complete precursor in any of these species.

Figure 1 Sequences of the last coding exons of the Mantodea EFLamide orthologs.

The conceptual translation of the last coding exon of the ELLamide neuropeptide precursor from six mantis species and the German cockroach are aligned. The two points before the first amino acid indicate the phase 2 intron site present in each of these exons. Blueish gray indicate the mature peptide sequence, the single arginine and the lysine-arginine doublet that serve as convertase cleavage sites are in red and the glycine residue that will be converted to a C-terminal amide is in purple. Note the mantid mature peptide is predicted to be SLGSELLamide, while its cockroach ortholog is likely to be an N-terminally extended neuropeptide.

Whereas the RYamide precursor has typically one or two paracopies, in Tenodera the gene went amuck. Its RYamide gene potentially codes for an RYamide precursor containing 20 RYamide paracopies. Since the genome assembly was obtained from HiFi PacBio sequences, this is not a genome assembly error as individual sequences from SRR28416710 attest. However, this is not a general feature of the RYamide gene in mantises, as the RYamide precursors from Mantis, Deroplatys, Hymenopus and Metallyticus are all predicted to have well conserved RYamide precursors that contain only three RYamide paracopies (Fig. 2).

Figure 2 Mantodea RYamide precursors.

The predicted RYamide precursors from T. sinensis, M. violacea, D. truncata, M. religiosa and H. coronatus. In four out of five species, these precursors are well conserved, but the Tenodera gene contains a large increase in the number of coding exons predicting a precursor that could contain 20 RYamide paracopies if all putative exons get incorporated into the final mRNA.

Genome and transcriptome analyses have shown that the last common ancestor of insects and vertebrates already must have had three distinct insulin-like peptides (ilps). Two of which, gonadulin, a putative ortholog of Drosophila insulin-like peptide 8, and a Drosophila insulin-like peptide 7 (dilp7) ortholog appear to act through a G-protein coupled receptor (GPCR), while the third, insulin-like growth factor (IGF), uses a receptor tyrosine kinase (RTK) (Veenstra, 2020, 2021a). Genes coding gonadulin, IGF and a dilp7 ortholog are located next to one another on chromosome 1, and organized in a similar fashion as their orthologs in the German cockroach. Apart from these three arch-ilps, insects have a second type of ilp, likely derived from IGF, since it also acts through an RTK, the co-called sirps (short IGF-related peptides). T. sinensis has five sirp genes that are located next to one another on a different part of chromosome 1 (Fig. S3).

The sirps are the most variable insect ilps. Interestingly, as in cockroaches, there are two types of sirps, those that have three or four amino acid residues between cysteine residues 1 and 2 of the A-chain (Fig. S4). The T. sinensis sirp 3 gene is unusual, as it is predicted to contain eight rather than six cysteine residues and is thus expected to have four disulfide bridges. Its putative precursor is also much smaller, as it lacks both the sequence corresponding to the connecting peptide and its associated convertase cleavage sites (Fig. S3). This gene seems to be strongly expressed in the periphery (Table 1). However, its ortholog in T. angustipennis is remarkably different, as it has only six cysteine residues, although like the T. sinensis peptide, it lacks a connecting peptide and its associated convertage cleavage sites (Fig. S4). Orthologs of this unusual sirp were neither found in the genomic SRAs from either D. truncata, M. religiosa, H. coronatus or M. violacea, nor in the transcriptome SRAs from 53 additional Mantodea. This peptide thus appears to have a very recent origin and may be limited to a single species.

Table 1 Expression of Ilps in Tenodera sinensis.

SRA	Tissue	Total	Dilp7	Gon.	IGF	Sirp1	Sirp2	Sirp3	Sirp4	Sirp5	
SRR25098587	Hind leg	33,554,395	0	0	11	8	0	17,600	19	3,363	
SRR25098588	Middle leg	32,504,063	0	0	49	1	0	1,650	11	1,145	
SRR25098589	Foreleg	35,407,910	0	0	9	0	0	42	7	1,602	
SRR25098590	Abdomen	36,794,322	0	0	13	8	0	418	7	10,996	
SRR25098591	Thorax	34,525,700	0	0	28	5	0	36	0	5,550	
SRR25098593	Head	37,231,830	0	0	37	116	0	739	57	20,162	
SRR25098594	Eye	33,942,656	0	0	9	8	0	398	6	3,934	
Note:

The number of spots coding dilp7, gonadulin, IGF and the five sirps in transcriptome SRAs prepared from different body parts of T. sinensis. Note the abundant and ubiquitous expression of sirps 3 and 5, while sirp 1 appears to predominantly expressed in the head. Total refers to the total number of spots in each SRA. Gon., gonadulin.

With one exception the Mantodea sirps differ significantly from those of cockroaches (Fig. 3). The exception is mantis sirp 5, that has a remarkably similar sequence as the cockroach sirp, that has been called atirpin (Veenstra, 2023). Like atirpin, mantis sirp 5 is abundantly expressed and likely in many different tissues (Table 1).

Figure 3 Sequence similarity tree of cockroach and mantid sirps.

Four of the Tenodera sirps have orthologs in other Mantodea. The only mantis sirp that has a cockroach ortholog is sirp-5, as suggested by the approximate probability of the branch uniting cockroach atirpin and mantid sirp-5. This branch has been highlighted in yellow.

As most of the mantis neuropeptides are very similar to those of other Polyneoptera, the inability to find tryptopyrokinin or any other pyrokinin in the Tenodera genome assembly was surprising. Neuropeptide genes are sometimes difficult to identify due to mutations in their relatively small consensus core sequences, as was also the case for the Tenodera EFLamide ortholog. In order to establish that the two pyrokinin signaling systems are genuinely lacking in this species, the genomic T. sinensis SRA SRR25098594 was searched for the presence of their putative receptors. Using the pyrokinin and tryptopyrokinin receptors in the same way as those for the EFLamide receptor, only reads that code for the Tenodera periviscerokinin GPCR were identified (periviscerokinin is a peptide closely related to the (trypto)pyrokinins and in Eumetabola called Capa). The simultaneous absence of individual reads for both receptors and ligands must mean that both pyrokinin and tryptopyrokinin have been lost in T. sinensis. Results from genomic SRAs from four other mantis species, i.e., Metallyticus violacea, Deroplatys truncata, Hymenopus coronatus and Mantis religiosa, similarly identified only the periviscerokinin receptor, suggesting that pyrokinin receptors could be missing from all Mantodea. Surprisingly, tryptopyrokinin genes, or what remains of them, are still present and expressed in some mantises. While there is not a trace of such a gene in either M. religiosa or T. sinensis, such genes are present in M. violacea, H. coronatus and D. truncata. Nevertheless, the coding sequence of the Hymenopus gene contains several indels, showing that it can not produce an intact tryptopyrokinin precursor and must be a pseudogene (Fig. 4). Accordingly, not a single tryptopyrokinin read was found in any of the transcriptome SRAs from this species. On the other hand, the tryptopyrokinin genes in Metallyticus and Deraplatys appear to be intact. While there is no transcriptome data for M. violacea, such data from M. splendidus demonstrate expression of a tryptopyrokinin gene, while the Deraplatys gene is also expressed. Tryptopyrokinins are usually expressed in the suboesophageal ganglion, which is located in the head, but in Deroplatys, tryptopyrokinin is expressed exclusively in the thorax and not in the head or abdomen. As two other neuropeptides that are typically expressed in the suboeosphageal ganglion—SMYamide and inotocin—are found in the head but not in the thorax transcriptome; the expression of tryptopyrokinin in D. truncata appears to be significantly different from that in other insects (Table 2).

Figure 4 Predicted Mantodea tryptopyrokinin precursors.

The predicted tryptopyrokinin precursors from M. violacea and D. truncata both code for a number of tryptopyrokinins. Although in both species a few mutated paracopies are no longer biologically active, the large majority has perfectly conserved C-terminal sequences. The conceptual reconstruction of the H. coronatus sequence has a larger number of mutated copies, but more importantly the gene sequence contains five mutations that lead to frame shifts that make it impossible to produce such a precursor. A putative transcript from this gene would have a stop codon before the first tryptopyrokinin coding sequence and can thus not produce this neuropeptide. The very repetitive structure of these precursors makes it possible to identify the details of the different mutations that inactivated this gene. In yellow the predicted signal peptides, in blueish grey parts of the sequences in the mature neuropeptides, in red the arginine residues that serve as convertase cleavage sites and in magenta glycine residues that are transformed in C-terminal amides. Finally, in black the mutations in the Hymenopus gene.

Table 2 Expression of selected neuropeptides in different body parts of Deroplatys truncata.

SRA	Tissue	Total	TryptoPK	SMYa	Inotocin	PeriVK	Leucokinin	
SRR25068528	Hind leg	46,674,984	0	0	4	0	1	
SRR25068529	Middle leg	40,175,963	0	0	0	0	7	
SRR25068530	Foreleg	41,714,914	0	0	4	2	6	
SRR25068531	Abdomen	43,344,826	0	0	2	0	7	
SRR25068532	Thorax	54,079,288	1,622	0	0	1	24	
SRR25068533	Head	38,062,541	0	3	13	18	259	
SRR25068534	Eye	48,189,596	0	1	4	1	16	
Note:

The number of spots coding tryptopyrokinin, SMYamide, Inotocin, Periviscerokinin and Leucokinin in transcriptome SRAs prepared from different body parts of D. truncata show that tryptopyrokinin is abundantly expressed in the thorax, but not in the suboesophageal ganglion where SMYamide and inotocin are produced by major neuroendocrine cells. Note that the absence of periviscerokinin spots in the abdomen transcriptome SRA does not equate the absence of periviscerokinin expression in the abdomen. The relative mass of the ventral nerve cord in the abdomen of a female is very small and hence CNS specific mRNAs are easily missed. Total refers to the total number of spots in each SRA. PeriVK, periviscerokinin; SMYa, SMYamide; TryptoPK, tryptopyrokinin.

The surprising presence and expression of tryptopyrokinin in mantises led to the analysis of all other publicly available Mantodea SRAs for the presence of spots that might code for tryptopyrokinin or its receptor. As the tryptopyrokinin genes code for many paracopies, the chances of finding tryptopyrokinin coding reads even in relatively small genome SRAs should be significant. In 19 genome SRAs from 16 different species, such reads were found, but the numbers of tryptopyrokinin spots in each SRA are relatively small, indicating that in some of the other 23 species a tryptopyrokinin gene may have escaped detection (Table S1). In 54 transcriptome SRAs belonging to 51 additional species, tryptopyrokinin reads were found in 30; when present, there are usually significant numbers (Table S2). When tryptopyrokinin is expressed, it is easily detected by analyzing a transcriptome SRA. It seems thus likely that the majority of species in which the transcriptome SRAs has no tryptopyrokinin spots, do not express this gene.

All the Mantodea SRAs were also searched for the presence of reads with sequences for the periviscerokin and (trypto-)pyrokinin receptors. A periviscerokinin receptor can be easily identified in the genome assembly from T. sinensis, and is similarly present in the genome SRAs from M. religiosa, M. violacea, D. truncata and H. coronatus. Reads unambiguously corresponding to this receptor are also found in virtually all Mantodea genome SRAs, as well as in 31 out 53 species, for which a transcriptome SRA is available. That the frequency of such reads in transcriptome SRAs is much lower than in the genome SRAs, is explained by the relative scarcity of the periviscerokinin receptor mRNA. The Blattodea have three pyrokinin receptors, of which one, the pyrokinin-1 receptor, is specific for tryptopyrokinins (Cazzamali et al., 2005; Homma et al., 2006; Olsen et al., 2007; Paluzzi & O’Donnell, 2012). Of those three pyrokinin receptors, only the pyrokinin-1 receptor was detected in SRAs. Reads corresponding to this receptor were found in a transcriptome SRA from M. splendidus, as well as a genome SRA from Leptomantella albella (Tables S1 and S2). Although both M. violacea and M. splendidus are assigned to the Metallyticus genus, no pyrokinin-1 receptor sequence could be found in M. violacea, illustrating interesting differences between these two species. There are four spots in the genome SRA from L. albella that contain coding sequences for a pyrokinin-1 receptor. Three, covering part of the first coding exon, that yield a 128 amino acid residue sequence, and one for the second coding exon predicting a 50 amino acid residue sequence. Each of these sequences unambiguously identify as being orthologous to the pyrokinin-1 receptors (Fig. 5). In Leptomantella this may well represent a functional receptor since: (1) there are no indels in these partial sequences, (2) the homology with the Periplaneta pyrokinin-1 receptors is excellent, and (3) two different coding exons are represented.

Figure 5 Evidence for Mantodea Pyrokinin-1 receptors.

Sequence comparison of the Periplaneta pyrokinin-1 receptor with deduced partial sequences from Leptomantella albella and Metallyticus splendidus. When used as query in a tblastn searching insect non-redundant protein sequences, all three sequences return pyrokinin-1 receptors as the most similar. Deduced sequences are from SRR18231900.8179607.1, SRR18231900.8179607.2, SRR18231900.2120256.1 and SRR18231900.4220800.1 (Leptomantella albella) and SRR921620.4158298.2 (Metallyticus splendidus).

In order to visualize the presence of tryptopyrokinin and its receptor in the Mantodea, a phylogenetic tree was made. A recent revision of the Mantodea phylogeny has shown that several families in this group are polyphyletic (Ma et al., 2023). It was therefore difficult to assess the correct phylogenetic position of quite a few species and these could thus not be included in the tree (Fig. 6).

Figure 6 Mantodea phylogenetic tree.

Indicated are the phylogenetic relations between various species analyzed here. Thick solid green lines indicate expressed tryptopyrokinin genes, broken thick solid green lines indicates species with tryptopyrokinin coding sequences. Thin lines indicate species for which no tryptopyrokinin reads were found in either a transcriptome or a genome SRA. The species for which this concerns a genome SRA are indicated with an asterisk. The two species for which partial evidence for a pyrokinin-1 receptor was recovered have been indicated with an arrow. Highlighted in black are the five species for which sufficient genome DNA sequences is available to determine unambiguously whether they have a tryptopyrokinin gene. Mantis and Tenodera do not have such a gene, while the other three do, although in the case of Hymenopus this is a pseudogene. Species for which the exact position on the phylogenetic tree was not clear could not be added to the tree. This tree is that recently published (Ma et al., 2023), from which genera without information on tryptopyrokinin have been removed.

Discussion

The Tenodera genome reveals a neuropeptidome quite similar to those found in cockroaches (Zeng et al., 2021; Veenstra, 2023), but it also has a number of unique features. Two of these, the unusual sirp that has eight cysteine residues and the enlarged RYamide gene, were not found in other mantises, and are likely to be only present in a relatively small number of species. Since the unusual sirp seems to be widely expressed, it is an interesting and intriguing peptide, but speculations with regard to its function would be premature. These results illustrate that the neuropeptidome of a single species is not necessarily representative for a whole insect order, as previously observed for Coleoptera (Veenstra, 2019).

In mantises, the well conserved C-terminal sequence of the EFLamides has mutated into ELLamide. The EFLamide to ELLamide transition likely happened before the separation of cockroaches and mantises, as the same sequence is also found in Blattella. The axonal morphology of the single bilateral EFLamide neuron in the migratory locust suggests its involvement in the modulation of spatial orientation (Veenstra & Šimo, 2020). It would be interesting to know whether ELLamide neurons have a similar morphology in praying mantises, as they are well known to have excellent vision and the possibility to perceive depth (e.g., Rossel, 1983).

Insulin- and IGF-related peptides have been extensively studied in insects, particularly in holometabolous species such as Drosophila melanogaster and Bombyx mori (e.g., Mizoguchi & Okamoto, 2013; Nässel & Vanden Broeck, 2016), but these peptides have received much less attention in Polyneoptera. The Tenodera genome assembly contains eight genes coding ilps, that are similar to those in Blattodea. It is interesting to note that both the sequence and the expression of Tenodera sirp 5 is very similar to cockroach atirpin, a sirp that has been suggested to be an autocrine, that is released when a tissue faces metabolic stress (Veenstra, 2023). It is tempting to speculate that Tenodera sirp 5 may have a similar function.

The presence of a tryptopyrokinin gene in species that have lost the tryptopyrokinin receptor raises the question whether a pyrokinin might also persist in some species. Given the relatively small pyrokinin consensus sequences, it is impossible to exclude the possibility that such genes exist, even though that seems to be unlikely.

The most striking feature of mantis neuropeptidomes is the absence of the pyrokinin receptors. Although a tryptopyrokinin precursor exists and is expressed in several Mantodea species, plausible evidence for a pyrokinin receptor was only found in two species. In both cases, this concerns the pyrokinin-1 receptor, the one that is specific for tryptopyrokinins (Cazzamali et al., 2005; Homma et al., 2006; Olsen et al., 2007; Paluzzi & O’Donnell, 2012; Jiang et al., 2014). It seems illogical that species may have a tryptopyrokinin gene but lack a specific tryptopyrokinin receptor. However, (trypto)pyrokinins and periviscerokinins are related peptides, and in some species the periviscerokinin receptors can also be activated by tryptopyrokinin, albeit in (much) higher concentrations than needed for periviscerokinins (Paluzzi et al., 2010; Jiang et al., 2014; Thakur, Park & Jindal, 2024). So perhaps in Mantodea the periviscerokinin receptors have a higher affinity for tryptopyrokinins. The large number of neuropeptide paracopies of the Mantodea tryptopyrokinin precursors might compensate to some extent for a lower affinity.

However, it is not only the absence of tryptopyrokinin that surprises, but also the unusual site of expression of tryptopyrokinin in D. truncata. In D. truncata, the tryptopyrokinin gene seems to be expressed in significant quantities in the thorax and likely not or hardly at all in the suboesophageal ganglion, its typical site of expression in other insect species. The expression of tryptopyrokinin in the thorax is remarkable and begs the question whether or not its function has also changed. Unfortunately, we don’t know whether this unusual expression is the rule or the exception in Mantodea. In many species tryptopyrokinins are also produced by the abdominal ganglia, but sensitive mass spectrometry methods failed to detect tryptoyrokinins in the abdominal ganglia in any of the forty Mantodea species tested, including Deroplatys desiccata (Koehler & Predel, 2010).

The data show convincingly that the tryptopyrokinin gene was lost in Mantis and Tenodera, and the phylogenetic tree suggests that this may have been a single loss event in all Mantidae (Fig. 6). In H. coronatus, where the peptide is no longer physiologically active, a second independent loss is in progress. Other species for which smaller transcriptome SRAs were analyzed and which yielded no tryptopyrokinin spots could still contain, and even express, a tryptopyrokinin gene. However, given the relatively large numbers of tryptopyrokinin spots generally found in Mantodea transcriptome SRAs, it seems likely that several of them have also lost a functional tryptopyrokinin gene. Due to their ambiguous phylogenetic position, many species could not be placed on the phylogenetic tree, but several of these would be placed somewhere between the Mantidae and Metallyticus, and hence would reveal additional independent tryptopyrokinin loss events. Clearly the loss of the tryptopyrokinin gene has happened multiple times.

Although it is possible that it represents a pseudogene, the apparent pristine condition of the partial pyrokinin-1 receptor sequences in Leptomantella must mean that at least at the origin of this clade, about 125 MYA (Ma et al., 2023), it must have been functional. It implies that this receptor was lost on at least two occasions, as pyrokinin-1 receptor sequencens were only found in two species. The low frequency of pyrokinin-1 receptor reads in transcriptome SRAs, as compared to the periviscerokinin receptor, may be due to a lower expression. However, these receptors are about the same size, yet the spots for the pyrokinin-1 one were only found in a single genome SRA, whereas those for the periviscerokinin receptor were found in nineteen. This suggests two things: (1) that like the ligand gene, the receptor gene was lost multiple times and (2) that the gene for the receptor was lost more rapidly than the one for the ligand. That the receptor may be more easily lost than the ligand is also suggested by its absence in three species—M.violacea, D. truncata and H. coronatus—that still have the tryptopyrokinin gene but have already lost the receptor. I have previously argued that since genes coding neuropeptide receptors tend to be much larger than those coding their ligands, the chance that chromosomal breaks lead to the destruction of neuropeptide signaling system is more likely to happen in a gene coding a receptor than one coding its ligand (Veenstra, 2019). This might explain why the receptor was lost more readily.

The most interesting question is: How is it possible that both pyrokinin and tryptopyrokinin signaling were lost from praying mantises? What do we know about the physiological function of these peptides that could explain why mantises are the only insect species that do not need these peptides? The repeated loss of these neuropeptide signaling systems even seems to suggest that mantises might be better off without them.

An early arthropod likely had a gene that coded for a pyrokinin/periviscerokinin-like peptide that subsequently evolved in one, coding both pyrokinin and periviscerokinin ligands. Such genes are present in ticks (e.g., Christie, 2008), while decapods and insects have separate genes for pyrokinins and periviscerokinins (e.g., Veenstra, 2016). Insects have a third ligand, tryptopyrokinin, that is already present in silverfish (Diesner et al., 2021). In Eumetabola, it is encoded by either or both of the pyrokinin and periviscerokinin genes, but the Polyneoptera evolved one or more specific genes that each encode multiple tryptopyrokinin paracopies. Although tryptopyrokinins are expressed in different cell types, two neuroendocrine cells in the labial neuromere of the suboesophageal ganglion in both Polyneoptera and Eumetabola typically produce almost exclusively tryptopyrokinins and are the main site of expression. In Locusta, that has three specific tryptopyrokinin genes, two of those are expressed there (Redeker et al., 2017). It is striking that these two orthologous neuroendocrine cells in all insect species always specifically produce tryptopyrokinins, even though this is achieved by different mechanisms (Predel & Wegener, 2006; Neupert et al., 2009). It is this that makes it so surprising, that praying mantises have lost tryptopyrokinins.

The other pyrokinins were initially identified by their ability to stimulate gut contractions, but they also increase contractions of the aorta (Holman, Cook & Nachman, 1986; Schoofs et al., 1991; Wagner & Cook, 1993). Neuroendocrine cell clusters in the mandibular and maxillary neuromeres of the suboesophageal ganglion are the major source of these neuropeptides (Kean et al., 2002; Predel et al., 2007). Like the neuroendocrine cells producing tryptopyrokinin, those cells are also well conserved in insects (e.g., Veenstra, 1984). As both the neuropeptides and the neurons expressing them are conserved, it is reasonable to assume that this is also the case for their function. In Drosophila the pyrokinin gene is known as hugin (Meng et al., 2002), and both the gene and the neurons expressing this gene have been extensively studied. Those and other hugin neurons in the thoracic ganglia play essential roles in feeding and locomotion as integrators of external and internal sensory cues (Melcher & Pankratz, 2005; Hückesfeld, Peters & Pankratz, 2016; Schlegel et al., 2016; King et al., 2017; Oh et al., 2021; Schwarz et al., 2021). As the hugin gene is so specific to these cells, it is tempting to speculate that these functions depend on pyrokinin. However, the hugin neurons that have no neurohaemal release sites, but exclusively project within the central nervous system use acetyl choline as a co-transmitter (Schlegel et al., 2016). It seems possible that the FMRFamide expressing SE2 and/or SE3 neurons in Drosophila (Schneider et al., 1993) are a subset of the hugin neurons. So FMRFamide might be another co-transmitter. It is thus conceivable, that in the absence of pyrokinin such neurons might still be functional. The strongly conserved pyrokinin producing genes and their expression, together with the well documented functional relevance of the hugin neurons makes the absence of pyrokinin signaling in praying mantises as surprising as the loss of tryptopyrokinin.

In two species where the pyrokinin-1 receptor has been studied in some detail, it is strongly expressed in the salivary gland and this seems to be its main site of expression (Chintapalli, Wang & Dow, 2007; Yamanaka et al., 2008). Unfortunately, similar information is lacking for Polyneoptera (there is only a small salivary gland transcriptome from Coptotermes formosanus; it contains a single spot for the pyrokinin-1 receptor). The strong and specific expression of the pyrokinin-1 receptor in the salivary gland suggests that it is an important regulator of salivation. In cockroaches and locusts, salivation can be stimulated by well characterized aminergic and peptidergic neurons in the suboesophageal ganglion, that have been shown to stimulate both enzyme and water secretion and are active during salivation. Whereas dopamine, 5-hydroxytryptamine, gamma-aminobutyric acid (GABA) and SMYamide are delivered directly on or in close proximity of the salivary gland, the tryptopyrokinins are released into the general circulation (Schachtner & Bräunig, 1995; Just & Walz, 1996; Baumann et al., 2004; Watanabe & Mizunami, 2006; Rotte et al., 2009; Veenstra, 2021b). It is unlikely that a hormone released into the hemolymph would be able to acutely regulate salivation in the same fashion as these neurons. The effects of tryptopyrokinin on the salivary gland must thus be more subtle. Its release into the hemolymph also suggests that its action is likely not limited to the salivary gland, but of a more general nature. In many insect species periviscerokinins and (trypto)pyrokinins are produced simultaneously from the same precursors, indicating that these peptides act in concert in the same physiological context. The effects of periviscerokinins may thus be helpful in understanding what this function is. The first periviscerokinin from Lepidoptera was identified by its cardiaocceleratory effects (Huesmann et al., 1995). They are often referred to as diuretic hormones, and then called capa, for their effects on the Malpighian tubules in Drosophila melanogaster (Davies et al., 1995), but in Rhodnius and Aedes these peptides are anti-diuretic (Paluzzi & Orchard, 2006; Sajadi et al., 2020). This difference likely reflects the disparate diuretic needs after feeding in these species and indicates that their function is not the elimination of excess water. In Drosophila, the expression of the periviscerokinin receptor is very broad and not limited to the Malpighian tubules. It is also expressed in visceral muscle, enteroendocrine cells, the brain and still other tissues that are important for properly regulating digestion and the metabolic changes it induces (Koyama et al., 2021). Thus the pyrokinins, tryptopyrokinins and periviscerokinins promote, perhaps sequentially, different aspects of feeding, digestion and the physiological consequences of the subsequent increase in metabolism.

This leads to the hypothesis that in anticipation of actual feeding, the salivary glands are primed by tryptopyrokinins; such priming might consist of inducing the synthesis of salivary enzymes (a testable hypothesis). As circulating nutrient levels increase, the periviscerokinins activate the Malpighian tubules and other tissues to accommodate the physiological changes associated with digestion. The role of pyrokinins in this scheme and the reason they may have been lost is uncertain. One might speculate that pyrokinins serve to activate the muscles needed for feeding, like athletes activate their muscles when preparing to run.

If this hypothesis were correct, it could provide an explanation as to why tryptopyrokinins and other pyrokinins may have been lost in praying mantises. Fruit fly maggots, silk worm caterpillars, grass hoppers, and even bed bugs, can reasonably anticipate when they will start feeding since they are either on or approaching their food source. They can thus prime their digestive system using (trypto)pyrokinins. On the other hand, an ambush predator can not anticipate when it will actually start feeding until the very last moment when the prey has been captured. Praying mantises are not the only insect predators that might face this problem, but they seem to be able to handle prey that is much larger compared to their body size, e.g. dragon flies or ant lions. The larger the size of the average prey, the larger the time interval between consecutive meals, and the less productive the early activation of the salivary gland. Loosing the pyrokinin-1 receptor eliminates the untimely stimulation of the salivary gland, while the tryptopyrokinins may still be useful on the periviscerokinin receptor. Thus mantises that are feeding on smaller prey, or had ancestors that did so, may have maintained the tryptopyrokinin receptor. This may explain the uneven species distribution of tryptopyrokinin and its receptor within the Mantodea.

Once a praying mantis has its quarry in sight, it is too late to use hormones to prime the salivary gland or the muscles of the forelegs and gut. On the other hand, by not priming the muscles needed for moving and capturing its prey the mantis may enhance its ability to keep completely still while laying in ambush. Once a prey is captured, it will take some time before nutrients enter the hemolymph—the postulated signal for the release of periviscerokinins into the hemolymph (Koyama et al., 2021)—so periviscerokinins remain useful. Furthermore, keeping the complete digestive system, gut and salivary glands activated while waiting for a prey to arrive might be energetically costly, perhaps even more so for a species that may have to wait a long time before it may feed again.

Conclusion

If pyrokinins are hormones released in anticipation of feeding and digestion, it could make sense that they are lost in praying mantis, as ambush predators can not anticipate their next meal.

Supplemental Information

Supplemental Information 1 Tenodera Neuropeptide Sequences.

The deduced coding sequences and their translations of the predicted neuropeptides from Tenodera sinensis.

Supplemental Information 2 Supplementary figures and tables.

I gratefully acknowledge all those who make their DNA sequences publicly available, without them this work would have been impossible. I thank Paul Taghert, Lapo Ragionieri and an anonymous reviewer for their constructive criticism.

Additional Information and Declarations

Competing Interests

The author declares that he has no competing interests.

Author Contributions

Jan A. Veenstra conceived and designed the experiments, performed the experiments, analyzed the data, prepared figures and/or tables, authored and reviewed drafts of the article, and approved the final draft.

Data Availability

The following information was supplied regarding data availability:

The sequences are available at NCBI: SRR25309874, SRR28416710, SRR25098587–SRR25098591, SRR25098593, SRR25098594, SRR25249010–SRR25249024, SRR25068526, SRR25046609, SRR25010894, SRR25078554.

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
