# Peer review of "Neuropeptides from a praying mantis: what the loss of pyrokinins and tryptopyrokinins suggests about the endocrine functions of these peptides"

_PeerJ, doi:10.7717/peerj.19036_

## Round 0.1 · original submission · Major Revisions

Dear Professor Veenstra,

Two out of three reviewers have suggested major revisions to your article. Please consider their comments in revising your manuscript.

Reviewer 1 ·

Basic reporting

.

Experimental design

.

Validity of the findings

.

Additional comments

PeerJ review
This paper uses genomic and transcriptomic data gleaned from public databases to assess the presence or absence of pyrokinin neuropeptides within species of the suborder Mantodea. This class of neuropeptides is nearly uqibuitous within the Arthropoda and contains individual neuropeptides that play several roles including regulation of pheromone biosynthesis, water balance in the gut, visceral muscle contraction and regulation of feeding behavior. Given the diversity of functions, it is unusual for pyrokinin genes and neuropeptides to be lost over evolutionary time, however, this is what the author has found in species of Mantodea. Beginning with the recently sequenced genome of one species, the author surveys genomic and transcriptomic DNA in multiple species to reveal that related peptides, the tryptopyrokinins and periviscerokinins, are retained in most (but not all) mantid species. Thus the function of the later groups are likely conserved while the functions of pyrokinins are lost, presumably because they are no longer adaptive and useful to these species.

The findings are interesting and contribute to knowledge of neuropeptide systems evolution, as evolutionary variation in presence or absence of the insect pyrokinin peptides is well known but deeper analysis is lacking. Even less is known about variation and evolutionary history of the pyrokinin GPCRs, which is also covered in the present paper.

The primary issue that I have with the manuscript is that I wonder whether “presence of absence does not equal absence of presence” applies to the data reported here. The author states as much in the Discussion (Lines 291-292). Can the author be certain that the lack of a particular gene in a genome or SRA truly means that it has been evolutionarily lost, or could it have not been recovered in the analysis? If the author has methodology or another rationale for concluding that missing genes are not artifactual, it should be included in the paper.

The Discussion seemed difficult to follow, and the hypotheses presented seemed at times implausible. I am confused by the rationale behind statements such as that found in Discussion Lines 292-295, which seems to suggest that since Mantodea overall have many tryptoPK hits in SRAs, it’s likely that many have also lost a tryptoPK gene. I would recommend that the author re-evaluate the content in the Discussion, and draw up a better supported and plausible hypothesis for observed patterns of gene presence or absence and the functional and the adaptive significance of the loss of pyrokinins in mantises.

Smaller comments are as follows:
The title suggests that the main topic is pyrokinins. Several other neuropeptides are covered in some detail as well. The title should be modified to reflect this.
Line 70- should be “So far”
Lines 72-73- It sounds like many resources previously existed for use in this study, not just the T. sinensis genome. To which extent did the newly sequenced genome facilitate this study? It seems that it was component of the search process rather than the foundation. Genome, SRAs, T. augustipennis SRAs
Line 87- “as query” can be better phrased?
Line 101- “to complete” should be “completion of”
Line 110- “rationale”
Lines 112-113- how do we know it’s statistically impossible? Maybe a different phrase like “becomes highly unlikely” would be more accurate.
Line 113- “receptors”
Line 193- “pseudogene”
Line 207-9- unclear. Whose “numbers are relatively small?” SRAs? Species? Trypto-PKs?
Lines 211-212- How did the author come to this conclusion?

Discussion
Line 76- Does the author mean “tryptopyrokinins” instead of periviscerokinins?
Lines 279-281- I disagree that expression of trypoPK in thorax rules out “by deduction” expression in the SEG. What evidence exists suggesting that this would be the case?
The hypothesis presented beginning in line 316 does is focused on PVKs and PK1/H peptides, not considering the apparent loss of the FXPRLamide PK2 pyrokinins at all. Drosophila hugin is invoked as part of the explanation for the mantid pattern of gene absence; hugin is a PK2 pyrokinin and it seems like comparing this peptide functionally with PVKs and PK1s may not be entirely accurate. These peptides are well characterized in Polyneoptera, including Blattodea so their loss is as interesting as that of PK1 and PVKs. Does the author have a hypothesis for how losing pyrokinins may have been adaptive specifically to the Mantodea that could be added to the Discussion?

388-390- there is no evidence that any of these peptides work directly on skeletal or visceral muscles in an endocrine manner. It is unclear what “priming” of these muscles is supposed to mean. Overall I don’t find this hypothesis to be plausible. I don’t think it is needed for this paper if the author goes into a bit more detail regarding the straightforward findings of the paper including PK2s.

·

Basic reporting

Clear, unambiguous, professional English language used throughout.
Yes

Intro & background to show context.
Not enough – see comment #1 (below)

Literature well referenced & relevant.
Yes

Structure conforms to PeerJ standards, discipline norm, or improved for clarity.
I defer to the Editors regarding the PeerJ standards

Experimental design

Figures are relevant, high quality, well labelled & described.
Yes

Raw data supplied (see PeerJ policy).
Yes

Original primary research within Scope of the journal.
I believe so

Research question well defined, relevant & meaningful. It is stated how the research fills an identified knowledge gap.
Yes

Validity of the findings

Impact and novelty is not assessed.
No, it is

Meaningful replication encouraged where rationale & benefit to literature is clearly stated.
Yes

All underlying data have been provided; they are robust, statistically sound, & controlled.
Yes

Conclusions are well stated, linked to original research question & limited to supporting results.
Yes but more would help – see comment #2 (below)

Additional comments

This is an interesting and well-researched scientific report on a problem in comparative genomic analysis. It addresses the representation of genes that encodes neuropeptide and neuropeptide receptors in a specific insect, the preying mantis.

The number of sequenced insect genomes is now large and still growing. This is positive because insect diversity presents a useful landscape with which to take stock of evolutionary trends. Such trends can confirm prior assumptions and conclusion (e.g., the substantial conservation of the genes in question, across numerous diverse insect species, confirms their suspected functional importance). In addition, the differences between different insect groups can be likewise illuminating and novel, and thereby offer a basis to update our consensus interpretations. This manuscript does exactly that: it cogently highlights the broad conservation of neuropeptide and neuropeptide receptor genes in the preying mantis. It goes on to argue on good evidence that a specific class of neuropeptides and their receptors has been lost. This is the basis for a concise and reasonable speculation concerning the function of that specific signaling system. The author argues that the loss of those specific genes may be explained by the lifestyle of the mantis (ambush hunter). I have only a few comments//suggestions to possibly help improve this very excellent manuscript.

Comment #1. I suggest the “typical” reader would benefit by a more dedicated and detailed introduction to the clade that includes the highlighted insect, the preying mantis. To illustrate what I consider a possibly “too-casual” approach to introducing the main character, I note that in the Abstract, two different mantis species are mentioned. However, we do not have the benefit of an over-all context in which to (easily) slot observations about individual mantis species, or comparisons to non-mantid ones. The third para of the Introduction does mention some evolutionary generalities, but only in a way that assumes a lot of a priori knowledge on the part of the reader:
“The insect order of the Mantodea is closely related to the Blattodea, yet their ecological
niche is markedly different. While termites and other cockroaches are scavengers, mantises are predators,…” [lines 61-63]
Are all Mantodea mantises? Are all mantises predators? Is Mantodea closely related to any other orders? An over-all schema, that illustrates the features the author thinks are critical, would very useful at an early stage in the manuscript. For example, this could be placed in the description of Figure 1 in the Results, or elsewhere (like the Introduction).

Comment #2. The manuscript offers the interesting hypothesis that the ambush predator lifestyle of the mantis is correlated with and likely explains the ability of this group to dispense with pyrokinin signaling (for reasons enumerated). There are other insects that likewise employ an ambush/ predator lifestyle (e.g., ant lions, ambush bugs. etc): they lie in wait for their prey with no a priori knowledge (except perhaps time of day, and changing weather conditions). Are there examples with sequences genomes? If not, at least some mention of the opportunity this new hypothesis for robust prediction (no pyrokinin genes) and hypothesis-testing

·

Basic reporting

The language is generally clear, but at times the author uses expressions that sound somewhat informal, like lab chat. For example: 'All SRAs used in this manuscript...' could be improved to 'All raw reads used...'

Nearly all major references are included in the paper, though I have added a few suggestions.

Some sections can be improved with additional information, and the raw data used for the publication, which were obtained from public databases, are all listed.

To fully support the hypothesis, I suggest that the author also include the quality control results for the selected data.

Experimental design

The paper aligns well with the aims of the journal, but I suggest more clearly defining the research questions for the readers. The investigation could be improved by verifying the quality of each raw dataset used in this study, as this information is currently lacking.

The presentation should also be improved, particularly regarding the strategy used for read quantification and the phylogenetic tree analysis

Validity of the findings

The novelty of the findings is significant, but the author should improve certain analyses. When referring to the quantification of RNA-seq reads, the author used a single RNA-seq per tissue. The issue here is the absence of biological replicates.

Additional comments

no additional comments

---

## Round 0.2 · Minor Revisions

Please address reviewer 2 and 3's comments, then resubmit the MS.

·

Basic reporting

Specific comments on metallic mantid species: First I quote from the text -
“Praying mantises belong to the insect order Mantodea. Although all Manodea are
76 matises and predators, not all mantises employ an ambush strategy; the metallic mantises
77 actively hunt and also use their forelegs not only to catch prey but also to move around,
78 which the other mantises do not."

“Reads corresponding
257 to this receptor were found in a transcriptome SRA from M. splendidus as well as a genome
258 SRA from Leptomantella albella (Tables S1 & S2). Although both M. violacea and M.
259 splendidus are assigned to the Metallyticus genus,…”

So then a Metallic mantis species is one of the two Mantid species (of 54 sequenced) that displays a pyrokinin receptor gene fragment – this supports the speculation/hypothesis that most ambushing Mantis species have lost the pyrokinin signaling pathway, but that a hunting Mantid species is one of two so far known to have retained it. Is my reading correct? I did not find this concisely stated as such (meaning stated “out loud”). If correct, I recommend this is be emphasized as a point of support for the hypothesis. One could do statistics on this, which many would support. I think the point is strong on the face of it, even without additional statistical analysis.

Regarding the peptide:
“When tryptopyrokinin is
243 expressed, it is easily detected by analyzing a transcriptome SRA. It seems thus likely that
244 the majority of species in which the transcriptome SRAs has no tryptopyrokinin spots, do
245 not express this gene.”

So are metallic mantis species among those for which there is evidence for retention of a tryptopyrokinin gene? Apologies if this information is included, but I missed it. [I could not access Table S2 - perhaps it is there?] In anycase, I think it should be prominently stated - "yes or no". Why? Because I agree with the author that this is issue (apparent loss/retention of a particular neuropeptide signaling system) is the main point of this paper. Such factors that weigh directly on how to interpret this conclusion should be presented to the reader in easily-captured fashion.

Experimental design

No comments here regarding the revision

Validity of the findings

Regarding the interesting discussions about absence/presence and whether the data indicate something biological about genomes or more technically how to collect data about genomes: I give the author the benefit of the doubt in choosing which conclusion to draw, as long as they do so explicitly, explain the technical considerations, and use the word speculation to describe inferences that are subject to opinion. All these things were done. The sum of which is to make for a thoughtful and provocative contribution. It may be that the author is correct that most species in this insect group lack this signaling system. Or it may be that the capture of genomic information for this group has a technical issue which precludes inclusion of these specific gene sets (is there precedent for that?). Regardless, I fully support this manuscript for taking a well-reasoned, novel and stimulating approach.

Additional comments

I think the author has responded carefully and thoughtfully to all the concerns and questions.

I have one recurring comment, regarding the presentation of data regarding metallic mantid species. As I understand it (summarized in #1 above), these are non-ambush Mantid species and at least one is one of the exceptions regarding retention of a pyrokinin GPCR. This and any information regarding cognate peptide genes should be concisely summarized as it represents information directly pertinent to the evolutionary-physiological speculation the author develops in the Discussion.

·

Basic reporting

I gave some suggestions how improve figures and tables.

Experimental design

no comment

Validity of the findings

The finding are interesting but the conclusion are not strongly supported.

Additional comments

no additional comments

---

## Round 0.3 · accepted · Accept

I have assessed the author's revisions and the comments of the reviewers. The paper is now ready for publication.

·

Basic reporting

no comment

Experimental design

no comment

Validity of the findings

no comment

Additional comments

no comment